# Mesenchymal-epithelial crosstalk shapes intestinal regionalisation via Wnt and Shh signalling

Martti Maimets [1,2,3,8], Marianne Terndrup Pedersen[1,2,8], Jordi Guiu[1,2], Jes Dreier[2,3,4], Malte Thodberg[1,2,5], Yasuko Antoku[1], Pawel J. Schweiger[1,2], Leonor Rib[1], Raul Bardini Bressan[1,2,3], Yi Miao [6], K. Christopher Garcia [6,7], Albin Sandelin [1,5], Palle Serup [2,3] & Kim B. Jensen [1,2,3✉]

Organs are anatomically compartmentalised to cater for specialised functions. In the small intestine (SI), regionalisation enables sequential processing of food and nutrient absorption. While several studies indicate the critical importance of non-epithelial cells during development and homeostasis, the extent to which these cells contribute to regionalisation during morphogenesis remains unexplored. Here, we identify a mesenchymal-epithelial crosstalk that shapes the developing SI during late morphogenesis. We find that subepithelial mesenchymal cells are characterised by gradients of factors supporting Wnt signalling and stimulate epithelial growth in vitro. Such a gradient impacts epithelial gene expression and regional villus formation along the anterior-posterior axis of the SI. Notably, we further provide evidence that Wnt signalling directly regulates epithelial expression of Sonic Hedgehog (SHH), which, in turn, acts on mesenchymal cells to drive villi formation. Taken together our results uncover a mechanistic link between Wnt and Hedgehog signalling across different cellular compartments that is central for anterior-posterior regionalisation and correct formation of the SI.

[1] Biotech Research and Innovation Centre (BRIC), University of Copenhagen, Copenhagen, Denmark. [2] The Novo Nordisk Foundation Center for Stem Cell Biology (DanStem), University of Copenhagen, Copenhagen, Denmark. [3] The Novo Nordisk Foundation Center for Stem Cell Medicine (reNEW), University of Copenhagen, Copenhagen, Denmark. [4] The Novo Nordisk Foundation Center for Protein Research, CPR, University of Copenhagen, Copenhagen, Denmark. [5] The Bioinformatics Centre, Department of Biology, University of Copenhagen, Copenhagen, Denmark. [6] Department of Molecular and Cellular Physiology, Howard Hughes Medical Institute, Stanford University School of Medicine, Stanford, CA 94305, USA. [7] Department of Structural Biology, Howard Hughes Medical Institute, Stanford University School of Medicine, Stanford, CA 94305, USA. [8] These authors contributed equally: Martti Maimets, Marianne Terndrup Pedersen. ✉email: kim.jensen@sund.ku.dk

The SI sub-regions - duodenum, jejunum and ileum - have characteristic cell types and tissue architecture that reflect their specialised functions in digestion and nutrient absorption. These regional traits emerge during late morphogenesis when a simple intestinal tube morphs into a more complex system with undulating structures. Although the structural and cellular changes during intestinal development have been extensively studied, the molecular determinants that support morphogenesis as well as establish and maintain small intestinal regional identity remain poorly understood[1–4].

Recent studies mostly performed in the adult tissues have highlighted the importance of stromal cell populations in providing structural support and producing growth factors necessary for crypt formation and maintenance[5–10]. Specifically, stromal cells marked by the combination of markers including Foxl1, CD34, PDGFRa, GLI1, and CD81 have been identified as important sources of WNT proteins and R-spondins central for regulating proliferation and differentiation. Wnt signalling impairment during morphogenesis leads to obvious developmental defects including impaired proliferation and a disruption in villus formation[11,12]. Notably, villus formation is a canonical example of the mesenchyme shaping structural changes in the intestine. Here, the Hedgehog (Hh) signaling pathway is activated in mesenchymal cells located beneath future villi leading to mesenchymal cell clustering and villus formation[13–15].

In this work, we hypothesised that specific mesenchymal cell populations contribute to regional specification of the developing SI along the antero-posterior axis. Therefore, we set out to characterise the basis for region-specific mesenchymal-epithelial crosstalk during late small intestinal morphogenesis. We show that a functionally distinct PDGFRa$^{high}$ mesenchymal cell population supports gradient activation of canonical Wnt signalling in the epithelium and uncover region-specific roles for the Wnt pathway in SI development. Notably, we demonstrate that expression of the epithelial Hh ligand, SHH, is directly regulated by Wnt signalling in the distal SI. Taken together our results reveal a new level of mesenchymal-epithelial crosstalk which promotes villus formation and shapes SI development.

## Results

### Fetal mesenchymal subpopulations show distinct and region-specific gene expression and subepithelial PDGFRa$^{high}$ cells support epithelial growth.

To gain insight into the molecular mechanisms that shape intestinal morphogenesis we set out to perform a detailed population-based characterisation of the epithelium and the underlying mesenchyme in the developing mouse SI at embryonic day (E)16.5. We first examined the literature for markers described as broadly expressed in intestinal mesenchymal cells. PDGFRa and CD29 stood out as candidates as PDGFRa$^+$ stromal cells are required for early intestinal morphogenesis, including the formation of intestinal villi[16], while CD29, marking all parenchymal cells, is required for correct compartmentalisation and regulation of intestinal proliferation[17]. Using 3D immunohistochemistry and flow cytometry, three mesenchymal cell populations could be distinguished based on the expression of CD29 and PDGFRa: (1) PDGFRa$^{high}$/CD29$^{med}$ (PDGFRa$^{high}$); (2) PDGFRa$^{med}$/CD29$^{med}$ (PDGFRa$^{med}$) and (3) PDGFRa$^{med}$/CD29$^{high}$ (CD29$^{high}$) (Fig. 1a, b and Supplementary Fig. 1a). We focused our further studies on the PDGFRa$^{high}$ population, due to its immediate vicinity to the developing epithelium, and the PDGFRa$^{med}$ population that represents a "bulk" mesenchymal signature. The CD29$^{high}$ population was not subjected to further analysis, as our results indicate that it represents the muscularis mucosa. Specifically, RNA-sequencing showed elevated expression of genes associated with muscle function such

as *Acta2*, *Myl4*, and *Des* (CD29$^{high}$ vs PDGFRa$^{med}$ (log2(fold change) > 0.5 and FDR < 0.05)) and gene ontology (GO) analysis displayed enrichment for genes involved in muscle system process and muscle contraction (Supplementary Fig. 1b, c and Supplementary Data 1 and 2). Furthermore, the CD29$^{high}$ population expressed elevated levels of LRIG1, which in the fetal intestine is highly expressed in the muscularis mucosa (Supplementary Fig. 1d, e).

In order to characterise PDGFRa$^{high}$ and PDGFRa$^{med}$ cells and to gauge their regional differences, we focused our attention on transcriptional profiling of these purified mesenchymal cells as well as epithelial EpCAM$^+$ cells from proximal, mid and distal regions of the E16.5 SI (Fig. 1c). Differential expression analysis detected 1342 genes with higher expression in PDGFRa$^{high}$ relative to PDGFRa$^{med}$ cells across all regions including *Lama5*, *Lamb1*, and *Npnt* (log2(fold change) > 0.5 and FDR < 0.05, Supplementary Fig. 2a and Supplementary Data 3). Immunostaining analyses confirmed that cognate proteins are prominently expressed by the subepithelial mesenchyme, which further supports the localisation of the PDGFRa$^{high}$ population (Fig. 1d). GO analysis of the common PDGFRa$^{high}$ signature showed an enrichment for genes involved in cell signalling and cell communication (Fig. 1e and Supplementary Data 4). In contrast, the 1021 genes up-regulated by the PDGFRa$^{med}$ population were enriched for DNA replication in the GO analysis (Supplementary Fig. 2b, c and Supplementary Data 4). Thus, the PDGFRa$^{high}$ and PDGFRa$^{med}$ cell populations have distinct characteristics and our analysis supports that the subepithelial PDGFRa$^{high}$ mesenchyme have unique signalling properties. Importantly, using in vitro organoid culture system[18,19], we observed that while both mesenchymal cell populations are able to support the growth of epithelial cells in the absence of essential growth factors (EGF, Noggin and Rspondin 1 – ENR), the PDGFRa$^{high}$ population has superior growth-promoting potential (Fig. 1f, g). These data demonstrate that subepithelial PDGFRa$^{high}$ mesenchymal cells secrete factors capable of supporting proliferation and expansion of the fetal epithelium.

Next, we compared gene expression patterns along the length of the SI (Supplementary Data 3). In line with previous studies, we observed that regional markers of SI epithelium were differentially expressed along the anterior-posterior axis including *Gata4*, *Onecut2* and *Osr2*[20,21] (Supplementary Fig. 2d–f). Interestingly, the PDGFRa$^{high}$ and PDGFRa$^{med}$ mesenchymal populations also showed region-specific expression patterns, with 560 and 212 genes differentially expressed along the length of the SI, respectively (Fig. 1h and Supplementary Fig. 2d, g). Taken together, we show that fetal subepithelial mesenchymal PDGFRa$^{high}$ cells are enriched for genes involved in cell signalling, supports fetal epithelial growth, and that mesenchymal SI cell populations, alike to the epithelium, show region-specific gene expression patterns at E16.5.

### Subepithelial PDGFRa$^{high}$ cells support an anterior-posterior Wnt signalling gradient.

Based on our results, we hypothesised that the PDGFRa$^{high}$ population plays a central role in paracrine signalling controlling intestinal regional identity. Therefore, we surveyed the RNA-sequencing data aiming to identify differentially expressed signalling ligands along the proximal-distal axis in this mesenchymal population (Fig. 2a)[22]. The analysis revealed a number of differentially expressed signalling molecules, including two canonical Wnt pathway agonists, *Rspo3* and *Rspo1*[23], with elevated expression in the distal segment (Fig. 2a and Supplementary Fig. 3a). Patterned expression of these ligands was further supported by in situ hybridisation (ISH) (Supplementary Fig. 3b). Conversely, the secreted Wnt antagonist *Sfrp1* and the two non-canonical *Wnt4*[24] and *Wnt16*[25] family members were

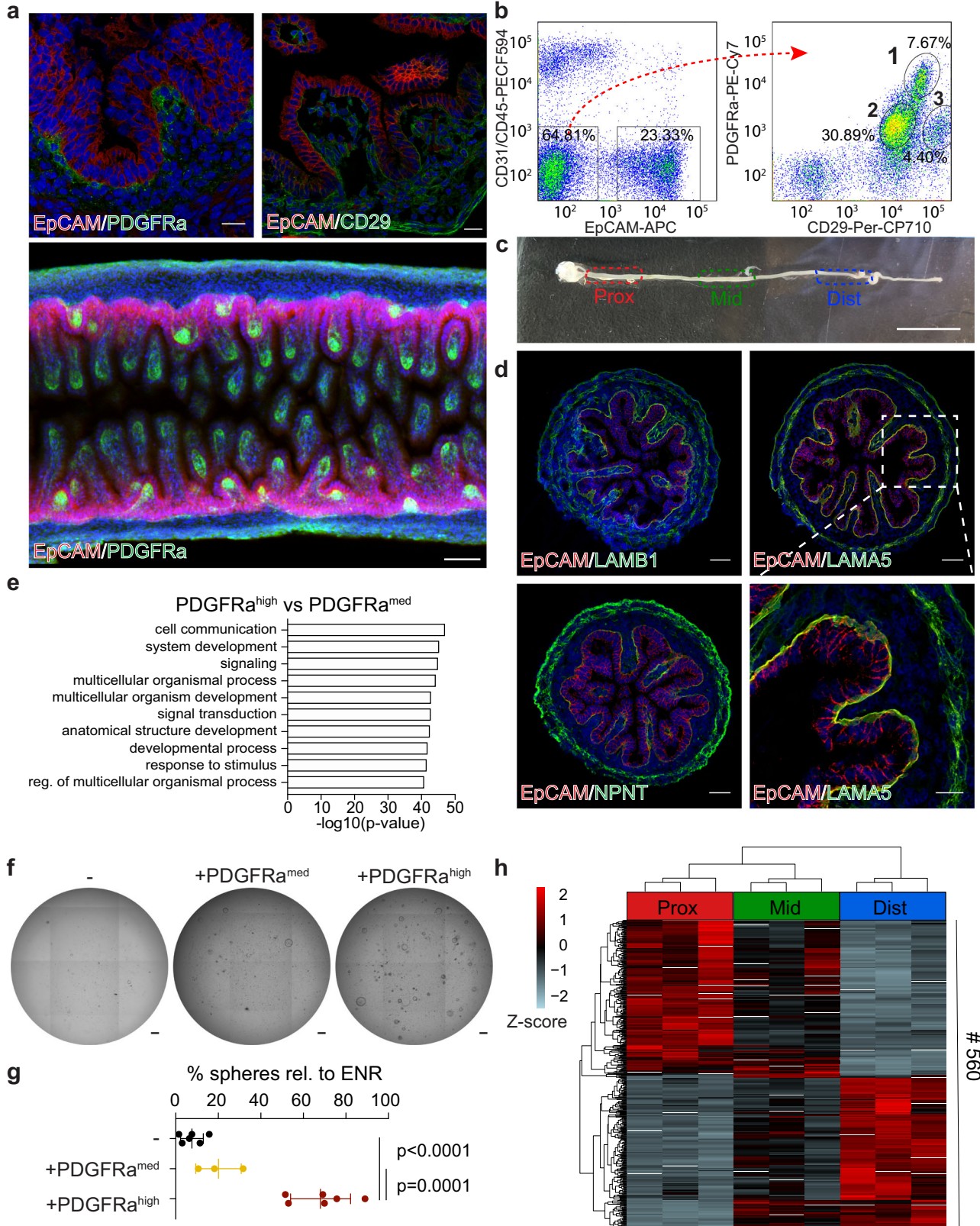

more highly expressed in the proximal segment (Fig. 2a and Supplementary 3c, d). Other Wnt ligands (Supplementary Fig. 3c) as well as most Frizzled receptors (Supplementary Fig. 3e) did not show significant region-specific expression patterns.

Given that non-canonical Wnts similar to sFRP1 inhibit canonical Wnt signalling[26,27], we hypothesised that the specific expression of antagonists and agonists might create the basis for a gradient of canonical Wnt signalling along the length of the intestine. In line with the potential gradient of Wnt activation and previous observations[12,28], we found the highest expression levels of two canonical Wnt target genes, *Axin2* and *Lgr5*, in the EpCAM+ cells isolated from the distal part of the SI (Fig. 2b). To

**Fig. 1 Subepithelial PDGFRa[high] cells support the growth of fetal epithelium and possess region-specific transcriptional profiles. a** Detection of EpCAM (red), DAPI (blue) and PDGFRa (green, left and lower panel) or CD29 (green, right) at E16.5. Lower panel is 3D-rendered image of a representative SI. Scale bars, 20 μm (upper panels) and 50 μm (lower panel). **b** FACS profile showing the distribution of DAPI⁻/CD31⁻/CD45⁻ parenchymal cells according to their EpCAM (left) and PDGFRa/CD29 (right) profile. **c** Overview of E16.5 SI and the division into proximal (prox), mid and distal (dist) regions. Scale bar, 1 mm. **d** Detection of EpCAM (red), DAPI (blue) and LAMB1 (green, upper left), LAMA5 (green, upper right and lower right), or NPNT (green, lower left) at E16.5. Scale bars, 50 μm except lower right, where scale bar, 20 μm. **e** GO-term enrichment analysis showing top 10 terms in the Biological Process category comparing PDGFRa[high] and PDGFRa[med] cells (intersect of pairwise tests as presented in Supplementary Fig. 2a). **f, g** Formation of organoids from DAPI⁻/CD31⁻/CD45⁻/EpCAM⁺ cells isolated from the proximal 20% of the E16.5 SI cultured with PDGFRa[med] (middle) or PDGFRa[high] cells (right) without ENR in basal medium. Percentage of spheres were calculated relative to organoid formation in basal medium with ENR. Scale bars, 100 μm. **h** RNA-seq of PDGFRa[high] cells depicting the 560 differentially expressed genes comparing proximal, mid and distal regions. Results in **a**–**d** and **f** are representative of at least $n = 3$ independent experiments. **h** presents RNA-seq data from $n = 3$ biological replicates collected in independent experiments. Data presented in **g** are means ± s.d. from $n = 6$ independent experiments for EpCAM⁺ and EpCAM⁺+PDGFRa[high] cultures and $n = 3$ independent experiments for EpCAM⁺+PDGFRa[med] cultures. Statistical significance was assessed using one-way ANOVA test. Figure e presents results from GO enrichment test using goana.

test this further, we characterised a transgenic mouse strain (Top.CFP)[29], in which CFP expression is under the control of multimerised TCF/LEF binding sites (Supplementary Fig. 4a). Confirming the specificity of the reporter, CFP expression was detected in sites of known Wnt activation at E16.5, including whiskers, eye lids, mammary glands and digits (Supplementary Fig. 4b) as well as in columnar shaped cells intermingled with Lyz1⁺ Paneth cells located at the bottom of crypts in the adult SI (Supplementary Fig. 4c)[30–34]. Interestingly, at E16.5 CFP expression was detected exclusively in the epithelial compartment of the SI and, in agreement with the transcriptional profiling data, the highest fraction of CFP positive cells was found in the distal fragment (Fig. 2c). We next leveraged the organoid culture system to interrogate whether epithelial cells display region-specific dependencies on WNT signalling. While epithelial cells from the proximal region grew efficiently in conventional culture medium (ENR) as described previously[19,35], cells from mid and distal regions required supplementation of Wnt surrogate (W)[36] for efficient organoid formation and expansion (Fig. 2d–f). Taken together these data show that PDGFRa[high] cells secrete Wnt signalling-associated molecules in a region-specific manner correlating with a proximal to distal Wnt activity gradient in the epithelium of the fetal SI. Moreover, epithelial cells derived from different segments of the developing SI have distinct Wnt-signalling requirements for in vitro expansion.

**Wnt signalling inhibition impacts gene expression and morphogenesis in a region-specific manner.** To investigate whether inhibition of Wnt signalling in vivo would have region-specific consequences for SI development, we treated pregnant dams twice daily from E12.5 to E16.5 with LGK-974[37], a well characterised PORCN inhibitor (PORCNi) that prevents secretion of Wnt ligands. Of note, this time frame marks the onset of villi formation in the developing SI. Importantly, we observed no overt adverse effects on pregnant dams nor on the overall viability of the embryos at E16.5. In agreement with Wnt loss-of-function phenotypes, PORCNi did however induce defects in limb and eyelids formation as well as the absence of hair follicles, which are three phenotypes that previously have been associated with loss of WNT signalling[38]. Furthermore, PORCNi treatment efficiently inhibited Top.CFP expression in E16.5 fetuses (Supplementary Fig. 5a) and reduced the number of Top.CFP positive cells in all SI segments (Fig. 3a and Supplementary Fig. 5b), confirming its inhibitory effect on canonical Wnt signalling. We therefore went on to assess transcriptomic and morphological changes caused by WNT inhibition in the epithelial cells. Indeed, RNA-seq analysis confirmed a pronounced downregulation of canonical Wnt/β-catenin target genes upon PORCNi treatment (Fig. 3b and Supplementary Fig. 5c). Strikingly, Wnt inhibition elicited the

greatest transcriptional response in the mid and distal segments (log2(fold change) > 0.5 and FDR < 0.05, Fig. 3c and Supplementary Data 5), in line with the observed differential activation of Wnt signalling along the proximal-distal axis. Morphologically, the length of the SI was significantly shorter, while the colon size remained largely unaltered upon Wnt inhibition (Supplementary Fig. 5d, e), which recapitulates genetic models of impaired Wnt signalling[12]. Tissue analyses of intestinal segments revealed a reduced but not abrogated cell proliferation as evident by EdU incorporation (Supplementary Fig. 5f). Interestingly, we observed a pronounced reduction in villus formation particularly in the distal segment (Fig. 3d). To quantify the morphological differences, we employed 3D microscopy followed by machine-learning based image analysis to segment the tissue into epithelium, villi, muscle and lumen (Fig. 3e, Supplementary Fig. 6a, b and Supplementary Movie 1). Focusing on villi, we found that PORCN inhibition led to a striking reduction in villus numbers along the SI, with the most prominent effect in the distal region, where villus formation was dramatically reduced (Fig. 3f–h). In contrast, villus length was not severely affected (Supplementary Fig. 6c), suggesting that, if successfully initiated, villus growth might progress in the absence of Wnt signalling. Finally, using the distance between epithelial and muscle layers as a readout of mesenchymal depth, we observed a marked decrease specifically in the distal region (Fig. 3i, Supplementary Fig. 6d and Supplementary Table 1). Collectively, this demonstrates that Wnt signalling plays a region-specific role in the developing SI with greatest impact on epithelial transcription and villus formation in the distal region.

**Wnt signalling controls Shh expression in the developing distal SI.** To gain further insights into the mechanism through which Wnt signalling affects SI morphogenesis, we sought to identify direct transcriptional targets of the pathway. To that end, we employed chromatin immunoprecipitation (ChIP) for TCF7L2 – the major co-factor for β-catenin and downstream transcriptional effector of Wnt signalling[11,39] – in EpCAM⁺ cells isolated from the E16.5 SI. We identified 6,238 regions bound by TCF7L2 including known TCF7L2 binding sites such as a region near the transcription start site (TSS) of *Axin2* (Fig. 4a, Supplementary Fig. 7a and Supplementary Data 6). Of the identified binding sites, 40.6% were located within 1 kb of an annotated TSS and 20.0% within gene bodies (Supplementary Fig. 7b). This distribution suggests that a high proportion of TCF7L2 binding sites are located at distal regulatory elements as also observed by others[40]. To identify potential WNT target genes responsible for villus formation, we reanalysed our RNA-sequencing data and focused on genes that were specifically upregulated in the distal relative to the proximal region, and which additionally were

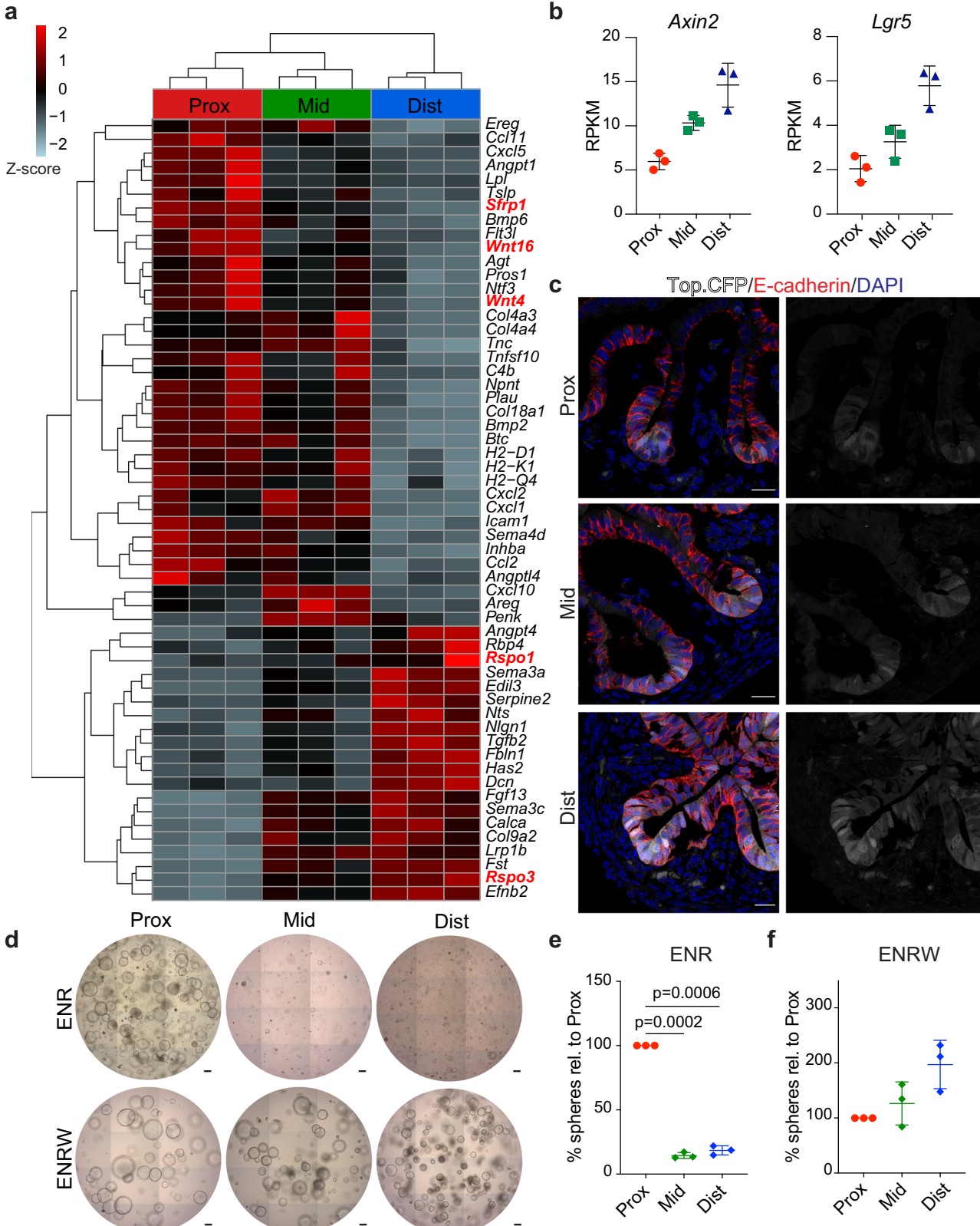

**Fig. 2 PDGFRa<sup>high</sup> cells-assisted Wnt-signalling gradient is present in E16.5 SI and required for the establishment of region-specific organoid cultures.**
**a** RNA-seq of PDGFRa<sup>high</sup> cells showing differentially expressed ligands between proximal, mid and distal regions. Highlighted in red are Wnt-pathway components. **b** RNA-seq results for *Axin2* and *Lgr5* from sorted EpCAM cells isolated from proximal, mid and distal regions. The y-axis shows RPKM (reads per kilobase per million mapped reads). **c** Detection of Top.CFP (white), E-cadherin (red) and DAPI (blue) in proximal, mid and distal regions. Scale bars, 20 μm. **d**–**f** Formation of organoids from DAPI⁻/CD31⁻/CD45⁻/EpCAM⁺ cells isolated from proximal, mid and distal regions and grown in the presence of ENR (**d**, **e**) or WENR (**d**, **f**). Scale bars, 100 μm. Results in **c** and **d** are representative of n = 3 independent experiments. In **b**, **e** and **f** data presented are means ± s.d. from n = 3 biologically independent samples. In **e** and **f** statistical significance was assessed using two-tailed Welch's *t*-test.

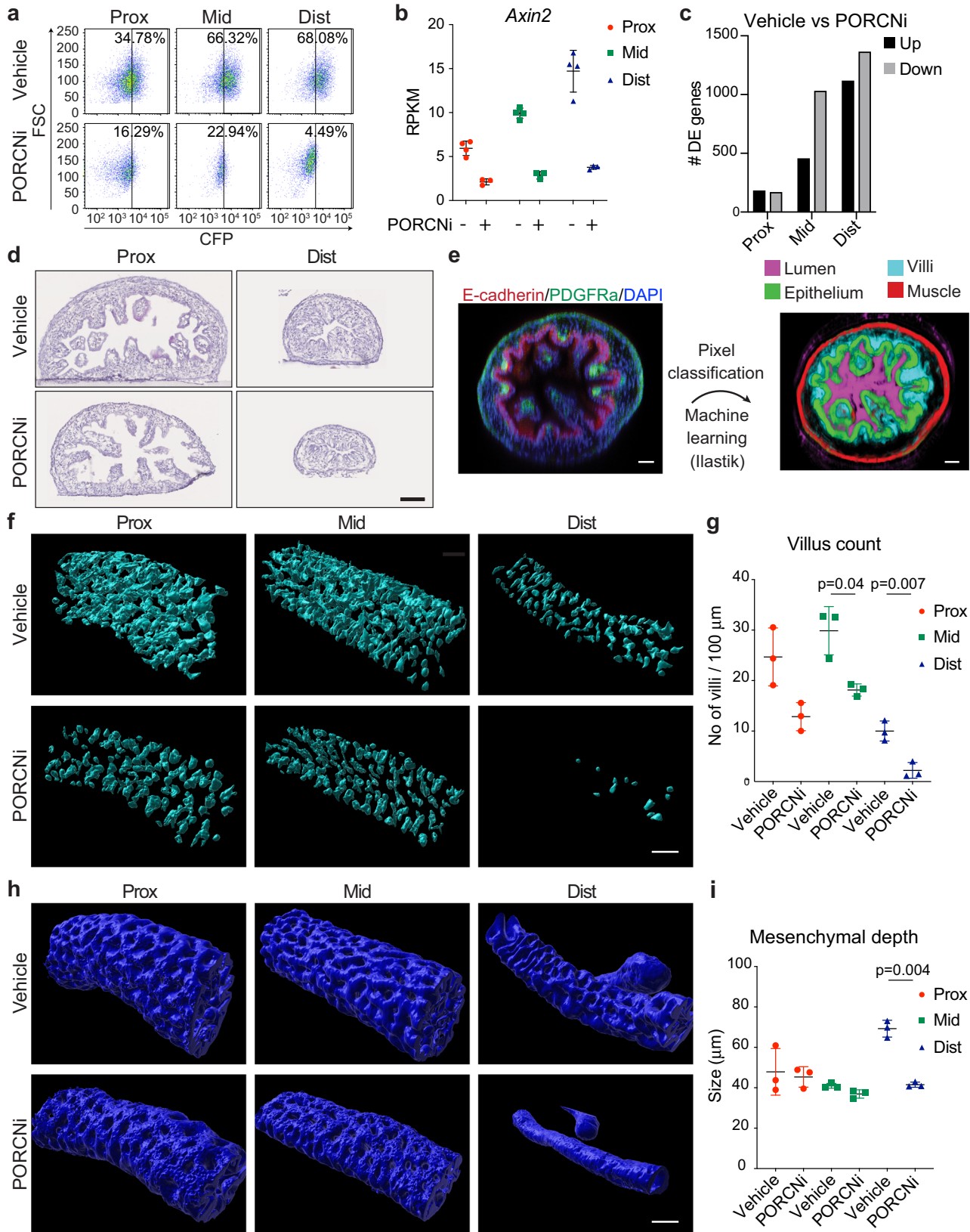

dependent on WNT signalling for high expression. This analysis identified 73 candidate target genes including the ligand for the Hedgehog pathway, *Shh* (Fig. 4b and Supplementary Table 2). Importantly, Hedgehog ligands expressed by the developing intestinal epithelium have been demonstrated to act on the underlying mesenchyme and promote villus formation[13]. In addition to SHH, the epithelium also expresses the hedgehog ligand, IHH (Indian hedgehog). However, while *Shh* was expressed in a gradient manner along the length of the E16.5 SI (Fig. 4c and Supplementary Fig. 7c), *Ihh* expression did not display this pattern nor was it affected by PORCN inhibition (Supplementary Fig. 7d). The gradient expression of *Shh* as well

**Fig. 3 Perturbation of Wnt-signalling has region-specific effects on SI transcription and structure. a** FACS profile showing the reduction of CFP+ cells (DAPI−/CD31−/CD45−/EpCAM+) in proximal, mid and distal regions upon PORCN inhibition (PORCNi). **b** RNA-seq data for *Axin2* with and without PORCN inhibition. **c** RNA-seq analysis showing the number of differentially expressed genes comparing control and PORCNi samples for proximal, mid and distal regions. **d** H&E staining of proximal and distal regions in control and PORCNi samples. Scale bars, 100 μm. **e** Workflow for machine learning-assisted 3D image analysis. Scale bars, 40 μm. **f** Surface renderings of Ilastik-processed probability maps depicting villi in control and PORCNi samples. Scale bars, 100 μm. **g** Quantification of villus count in proximal, mid and distal regions in control and PORCN inhibited samples. **h** Surface renderings of Ilastik-processed probability maps depicting epithelium in control and PORCNi samples. Scale bars, 100 μm. **i** Quantification of mesenchymal depth in proximal, mid and distal regions in control and PORCNi samples. Results in **a** and **d** are representative of $n = 3$ independent experiments. Results in **f** and **h** are representative of $n = 3$ biological replicates. Data presented in **b** are means ± s.d. from $n = 4$ (Ctrl) or $n = 3$ (PORCNi) biological replicates. In **g** and **i** data presented are means ± s.d. from $n = 3$ biological replicates and statistical significance was assessed using two-tailed Welch's *t*-test.

as the Wnt target genes, *Axin2* and *Lgr5*, was already present at E13.5 indicating that the differential expression did not relate to a temporal delay in villus formation along the antero-posterior axis (Supplementary Fig. 7e, f and Supplementary Data 7). Interestingly, inhibition of Hedgehog signalling has been demonstrated to block the initiation of villus formation[13,41] and reduce mesenchymal depth[15]. Thus, several of the effects observed upon PORCN inhibition in the distal SI (Fig. 3 and Supplementary Fig. 6) phenocopy impaired Hedgehog signalling.

Tissue-specific *Shh* expression is controlled by several enhancers. The so-called MACS1 enhancer region located within the intron of *Rnf32* has been shown to direct *Shh* expression in the developing intestinal epithelium[42]. Notably, our ChIP-seq analysis of E16.5 SI epithelium for TCF7L2 binding included a prominent peak at the MACS1 enhancer region (Fig. 4d). Transcriptional analyses of intestinal biopsies at E16.5 following vehicle or PORCN inhibitor treatment from E12.5 to E16.5, confirmed that *Shh* is most highly expressed in the distal SI and down-regulated upon inhibition of Wnt signalling (Fig. 4e). To address the effect of WNT signalling on *Shh* expression in a purely epithelial system, we derived organoids from E16.5 SI. Established organoid cultures were treated for 24 h with IWP-2, an alternative PORCNi, or IWR-1-endo, a tankyrase inhibitor, both blocking WNT signalling. Within this short time frame, there were no obvious effects on organoid growth or morphology, however, the high mRNA levels of *Shh*, *Axin2* and *Lgr5* observed especially in distal SI organoids were strongly reduced (Fig. 4f, Supplementary Fig. 7g). The downregulation of *Shh* upon WNT inhibition combined with the identification of TCF7L2 binding to an *Shh* enhancer, provides compelling evidence that *Shh* expression is directly regulated by WNT signalling in the developing distal SI.

To probe spatially the effect of PORCN inhibition on Hedgehog signalling, we performed ISH for *Ptch1*, a direct target of Hedgehog signalling and crucial receptor for Shh[43]. *Ptch1* expression was detected throughout the mesenchyme with a high concentration relatively close to *EpCAM*+ cells. In the distal SI, the distance between the epithelium and *Ptch1* mesenchymal expression was increased upon PORCN inhibition (Supplementary Fig. 7h, i). Interestingly, BMP4 has been reported as a transcriptional target of Hedgehog signalling in the intestinal mesenchyme[44]. Aligned with our proposed model, we observe a region-specific reduction in epithelial expression of the Bmp-responsive genes, *Id1* and *Id3*, upon PORCN inhibition (Supplementary Fig. 7j). These data are consistent with impaired secretion of Hedgehog ligands from the epithelium upon PORCN inhibition.

Lastly, we aimed to functionally address the relevance of the crosstalk between WNT and Shh signalling. For this, we utilised a mouse fetal whole intestine culture system that allows ex vivo development of villi[13]. To visualise mesenchymal cluster formation, a key requirement for villus genesis, we isolated E13.5 whole gut tubes from PDGFRa-CreER[T2] mice[45] crossed with

ROSA[mT/mG] animals[46] and cultured these in the presence of 4-hydroxy tamoxifen to enable continuous GFP-labelling of PDGFRa expressing mesenchymal cells. For 48 h the viability of the cultures was not affected as monitored by peristalsis and cell refractility (Supplementary Movie 2). As described previously[13], clusters of PDGFRa+ cells emerged within 48 h of culture. In line with our in vivo data (Fig. 3), formation of such clusters was strongly suppressed by PORCNi validating the use of the system to model villi formation (Fig. 4g, h). Strikingly, the effect of PORCNi could be fully rescued by coadminstration of a Smoothened agonist (SAG), demonstrating that Wnt signalling acts through Shh activation to regulate villi formation. Of note, activation of Shh signalling beyond control levels (SAG only) had little effect on the density of PDGFRa clusters (Fig. 4g, h).

In conclusion, our results uncover an important mesenchymal-epithelial crosstalk involving two central signalling pathways that shape the morphogenesis of the small intestine. We propose a model in which signalling molecules produced in distal subepithelial PDGFRa[high] mesenchymal cells support the activation of canonical WNT signalling in the epithelium directly stimulating expression of the Hedgehog ligand, SHH (Supplementary Fig. 7k). Hedgehog ligands secreted from the epithelium, in turn, act on the underlying mesenchyme to shape important morphological properties of the developing SI such as villus formation.

## Discussion

Key pathways are known to play critical roles during intestinal development, but how these patterning signals are coordinated through different cellular layers to achieve correct morphogenesis remains poorly understood. Mesenchymal cells in intestinal stroma drive epithelial fates during development as well as in tissue homeostasis[47–49]. However, the cellular identity of specific niche cells and their role in guiding epithelial regional specification have remained elusive. Here, we characterise PDGFRa[high] mesenchymal cells that during late intestinal morphogenesis constitute a cell layer that is juxtaposed to the epithelium throughout the small intestine. This is an ideal position to act as a niche, guiding patterning of the epithelium. Importantly, PDGFRa[high] cells have region-specific identities that are distinct from the remaining mesenchymal cells and express genes promoting a WNT signalling gradient. Whether the secretion of WNT associated molecules is robust in all PDGFRa[high] cells, or whether it is confined to specific cells with a unique position in the tissue, remains an open question. Also, our data does not rule out that other mesenchymal cells contribute to the observed patterns of WNT signalling and the source of canonical Wnt.

Our transcriptomic analysis of PDGFRa[high] cells reveals divergent expression of inhibitors and activators of the canonical WNT pathway including *Rspo1*, *Rspo3*, *Sfrp1*, *Wnt4* and *Wnt16*. Importantly, blocking Wnt signalling uncovers its region-specific role in villus formation especially in the distal region of the developing SI. Villus morphogenesis is a process where the flat

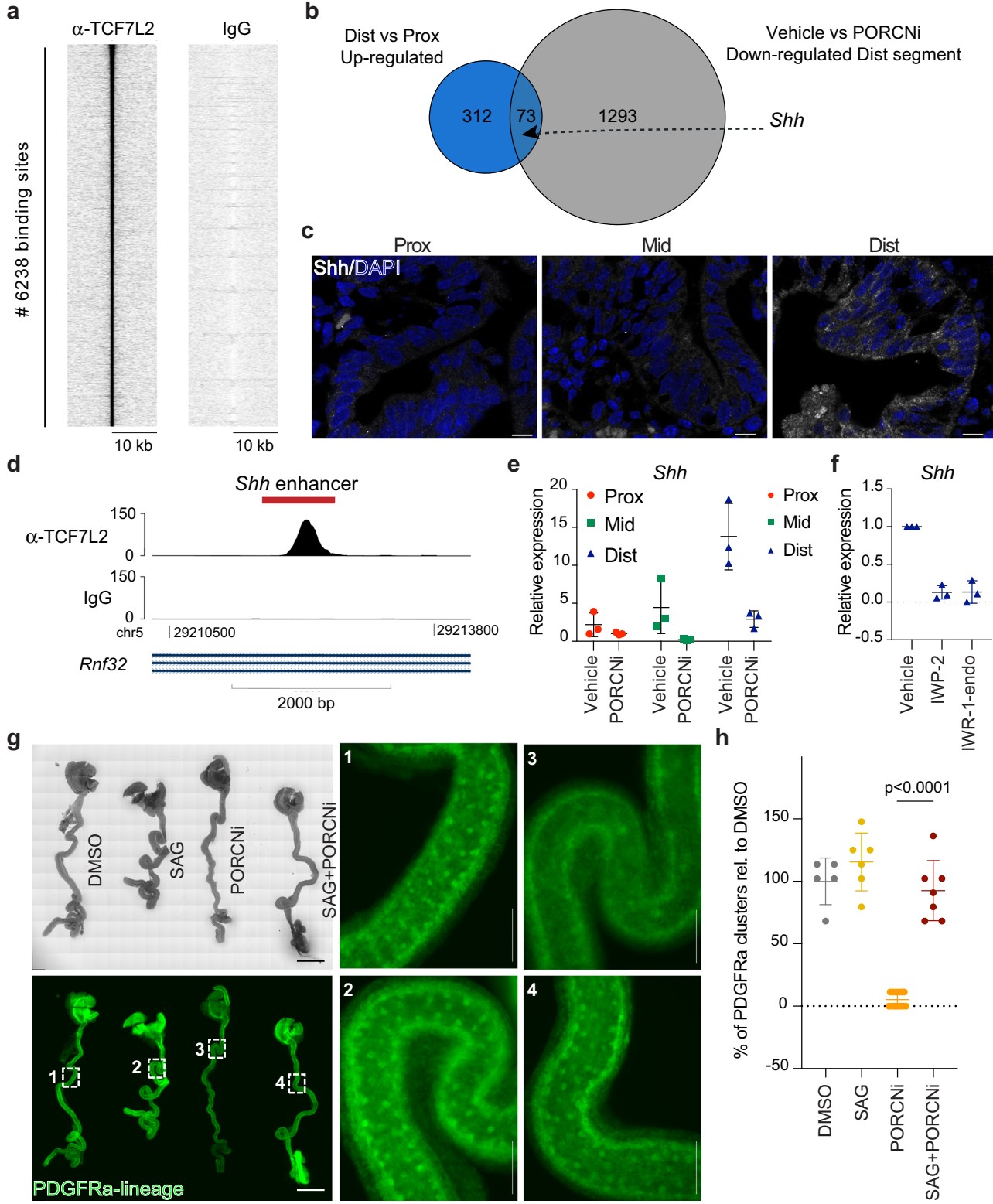

pseudostratified intestine begins to remodel and give rise to epithelial protrusions with an underlying mesenchymal core[15]. During villus formation, epithelial Hedgehog signals promote aggregation of subepithelial mesenchymal clusters that drive villus emergence[13]. Our data suggests a mechanism by which PDGFRa[high] mesenchymal cells act through activation of WNT signalling in the epithelium to stimulate expression of the Hedgehog ligand SHH.

WNT and Hedgehog signalling pathways are central players in both development and disease and complex examples of crosstalk between the two pathways have previously been identified in various cellular settings including tooth, hair follicle, mammary gland development as well as cancer development[50,51]. Thus, the observed direct regulation of Shh by WNT mediated TCF7L2/β-catenin signalling is likely to be used in multiple tissues.

**Fig. 4 Wnt signalling regulates Shh expression in the developing distal SI. a** Heatmap showing enrichment of ChIP-seq tags over the 6238 identified TCF7L2 binding sites (±10 kb). **b** Euler diagram showing the overlap between the 385 genes upregulated comparing distal versus proximal E16.5 SI EpCAM$^+$ cells (see also Supplementary Fig. 2d) and the 1366 genes down-regulated in control-treated relative to PORCN-inhibited distal regions (see also Fig. 3c). **c** Detection of SHH (white) and DAPI (blue) at E16.5. Scale bars, 20 µm. **d** ChIP-seq track depicting TCF7L2 coverage at the MACS1 Shh enhancer region (chr5:29211998-29212804, marked in red) in SI EpCAM$^+$ cells. **e** qPCR analysis for Shh using intestinal biopsies from E16.5 embryos. **f** qPCR analysis for Shh using established organoid cultures from E16.5 distal SI cultured in ENR and treated with Vehicle, 5 µM IWP-2, or 10 µM IWR-1-endo for 24 h. **g** Fetal whole intestine culture depicting cluster formation in PDGFRaCreER$^{T2}$;mTmG mice following DMSO (1), 1 µM SAG (2), 1 µM PORCNi (3) or SAG+PORCNi (4) treatment. Scale bars, 2 mm (overview) and 275 µm (enumerated panels). **h** Quantification of the number of PDGFRa$^+$ clusters. Results in e represent $n = 3$ biological replicates, and results in **f** represent $n = 3$ independent experiments. Results in **h** are derived from two independent litters of embryos where in total 4 (DMSO), 5 (SAG), 4 (PORCNi) and 6 (SAG+PORCNi) whole intestines were cultured. The data-points represent $n = 5$ (DMSO), $n = 6$ (SAG), $n = 13$ (PORCNi) and $n = 7$ (SAG+PORCNi) analysed images. Statistical significance was assessed using one-way ANOVA test. Graphs show means ± s.d.

## Methods

**Mice**. All animals were housed in specific pathogen-free animal facilities, in either open or individually ventilated cages always with companion mice, and cages were placed under a 12-h light–dark cycle. Food and water were provided ad libitum. Randomised cohorts including both male and female animals were distributed in an unblinded manner into the experimental time points for analysis. Sample sizes were selected to provide sufficient statistical power for analysis.

Unless otherwise specified, C57BL/6J mice (purchased from Taconic) were used for all the experiments. Top.CFP[29], PDGFRa-CreER$^{T2}$ [45] and ROSA$^{mT/mG}$ [46] murine lines used in the experiments have been described. The Porcupine inhibitor LGK974 (Selleckchem) was administered in a concentration of 10 mg/kg BID (oral) every 12 h for the duration of the experiment in a vehicle of 0.5% Tween-80/0.5% methylcellulose. None of the animals used in these studies had been subjected to prior procedures and all were drug- and test-naive. The Danish animal inspectorate reviewed and approved all animal procedures (Permit number 2017-15-0201-01381).

**Two-dimensional immunostaining**. Proximal, mid and distal segments of the SI were fixed with 4% paraformaldehyde (PFA) from 3 h to overnight at 4 °C, embedded in OCT and kept at −80 °C until sectioning. Sections (7 µm thick) were prepared with a cryotome (Cryostat CM 3050S).

Images from Hematoxyline QS (Vector) stained were acquired using NDP Zoomer Digital Pathology (Hamamatsu) and subsequently analysed in NDP.view2 software.

To perform immunofluorescence, sections were blocked and permeabilized in 10% adult bovine serum (Sigma), 5% skim milk (Sigma) and 0.3% Triton X-100 in PBS (blocking buffer) for at least 1 h at 4 °C. Primary antibodies (listed in Supplementary Table 3) were incubated overnight in blocking buffer at 4 °C. Alexa-Fluor-Conjugated secondary antibodies (indicated in Supplementary Table 3) were incubated for 1–2 h at room temperature in 10% adult bovine serum and 0.1% bovine serum albumin (BSA) (Sigma) in PBS. Diamidino-2-phenylindole dihydrochloride (DAPI; 1 µM; Sigma) was used to counterstain nuclei in the indicated experiments. Images were acquired using laser-scanning confocal microscopes (Leica TSC SP8 and Zeiss LSM800). All images were subsequently analysed with Fiji software.

**EdU staining**. Pregnant dams carrying embryos at E16.5 were intraperitoneally injected with 500 µg EdU and tissue was processed according to the manufacturer's instructions (Click-iT Plus EdU Alexa Fluor 488 Flow Cytometry Assay Kit, Thermo Fisher). Following a 1 h chase, the tissue was processed as described in the previous section. Before primary and secondary antibody staining, EdU was developed using Click-iT technology. DAPI (1 µM; Sigma) was used to counterstain nuclei in the indicated experiments.

**In situ hybridisation**. co-ISH was performed using RNAscope 2.5 Duplex Reagent Assay (ACD Biotechne). Briefly, tissues were fixed in 4% PFA overnight at room temperature before paraffin embedding. ISH was performed on deparaffinized and rehydrated 7 µm tissue sections according to the manufacturer's instructions using probes Mm-Sfrp1 (404981-C2), Mm-Rspo1 (401991-C2), Mm-Rspo3 (402011-C2), Mm-Wnt16 (401081-C2), Mm-Wnt4 (4011019), Mm-EpCAM (418151) and Mm-Ptch1 (402811-C2).

**Whole-mount immunostaining**. Proximal, mid and distal regions of the SI were fixed with 4% PFA from 3 h to overnight at 4 °C followed by dehydration in methanol. Samples were stored in methanol at −20 °C. The tissue was rehydrated in a PBS series before staining with all steps being performed at 4 °C. Samples were blocked and permeabilised in 1% BSA, 0.5% Triton X-100 (Triton X-100 Surfact-Amps Detergent Solution, Thermo Fisher) in PBS for 24 h. Primary antibodies (listed in Supplementary Table 3) were incubated in 1% BSA, 0.5% Triton X-100 in PBS for 48 h. The tissue was subsequently washed in 0.5% Triton X-100 in PBS overnight. Secondary antibodies (listed in Supplementary Table 3) were incubated in 1% BSA, 0.5% Triton X-100 in PBS for 48 h and the tissue was subsequently

washed overnight with 0.5% Triton X-100 in PBS. DAPI was used to counterstain nuclei in the indicated experiments. Samples were then dehydrated in methanol and kept at −20 °C. Samples were cleared using 1:2 benzyl alcohol:benzyl benzoate (BABB) (Sigma). In brief, 5 changes (1 min each) of BABB:methanol 1:1, then 5 changes (2 min each) of BABB. The samples were subsequently mounted within a Fast well (FW20-FastWells 20-mm diameter × 1.0-mm depth or 25 × 25 mm, Grace-Biolabs)[52]. Z-stack images were acquired using laser-scanning confocal microscopy (Leica TSC SP8).

### Image analysis

*Conversion of fluorescence images to morphological structures.* Following antibody staining for E-cadherin, PDGFRa and DAPI to segment epithelium from mesenchyme the three fluorescence channels were converted into physiological elements using a machine learning pixel classification in Ilastik[53]. Seven features maps were chosen for each channel resulting in 21 feature maps used in the classification. Sparse annotation was performed on each of the different proximal, mid and distal sections on both, vehicle and PORCN inhibitor-treated samples using the same classifier for all images used in this study. This resulted in a new 6 channel image, where each channel is a probability of the pixel belonging to either the background or one of the 5 morphological structures: lumen, muscle, villi, epithelium or mesenchyme layer.

*Segmentation.* The probability maps from ilastik were imported into Imaris (Bit-plane, Oxford, US) where the different elements were segmented. Lumen, muscle, epithelium and villi were segmented for visualisation and qualitative analysis. Subsequently, villi were manually processed in order to ensure each villus was an individual object, allowing us to perform quantitative measurements on the length and the number of villi. The length was defined as the longest ellipsoid axis, which could be contained within the villi, and was normalised to the length of the imaged SI. Events where a structure was below 20 µm were manually removed from quantification. The villi count was normalised to the length of the imaged section of the SI.

*Measurement of mesenchyme thickness.* The mesenchyme thickness was measured using the probability maps for the mesenchyme. The image was imported into Fiji, where a line was drawn perpendicular to the SI, using the re-slice function that generates a transversal section image. A line profile in the equatorial plane was used to measure the thickness.

*Colour deconvolution of two-dimensional immunostaining.* To separate EpCAM and Ptch1 ISH signals colour deconvolution was used in Fiji[54] using a matrix designed from our data. Prior to the colour deconvolution the images were background corrected. The two resulting channels were then inverted and a blue and a fire colour map was added respectively.

*Measurement of distance between Ptch1$^+$ and EpCAM$^+$ cluster.* On the images the epithelial border was marked as a continuous line. On this line, 10 points were drawn in a randomised single-blinded manner. From these points the distance to the closest Ptch1$^+$ cluster was measured.

**Isolation of epithelial and mesenchymal cells from the fetal small intestine by flow cytometry**. Small intestine from E16.5 or E13.5 fetuses was dissected and incubated with collagenase (Sigma) (125 µg/ml) in 0.1% BSA in PBS for 45 min at 37 °C and subjected to vigorous pipetting every 15 min using a P1000 pipette. Released cells were pelleted and resuspended in PBS supplemented with 1% BSA and incubated with fluorescent-conjugated primary antibodies (indicated in Supplementary Table 3) for 15 min at room temperature. After washing, 1 µM DAPI was added to the cell suspension to facilitate exclusion of dead cells by flow cytometry. Purified cell populations were subsequently isolated using a FACSAria III (BD Bioscience). FACS data were analysed using BD FACSDiva v9.0 (BD Bioscience) or Kaluza 2.1.2 (Beckman Coulter Life Sciences) software.

**Cell culture**. EpCAM⁺ cells were embedded in growth factor reduced Matrigel (Corning) in the presence of human EGF (Peprotech; 50 ng/ml), murine noggin (Peprotech; 100 ng/ml), human noggin (Peprotech; 100 ng/ml), mouse R-spondin1 (R&D Systems; 500 ng/ml) (ENR) in basal medium (Advanced DMEM/F12 (Gibco) supplemented with Pen/Strep (Gibco) and Glutamax (Gibco), N2 (Gibco), B27 (Gibco)). In co-culture experiments, 50.000 FACS-purified epithelial and 50.000 mesenchymal cells were mixed and embedded in growth factor reduced Matrigel (Corning). In some experiments surrogate Wnt[36] (10% in volume) was added as a supplement as indicated. For FAC-sorted cells, Y-27632 (Sigma) was added upon seeding. Cells were cultured in a dome-shaped 25-μl extracellular matrix droplet, and 250 μl culture medium was added to each well in a 48-well plate (flat bottom; Corning). Medium was subsequently changed every 2–3 days.

In order to generate organoid cultures from proximal, mid and distal parts of the E16.5 SI, intestinal fragments were incubated with Collagenase solution (125 μg/ml) in 0.1% BSA in PBS for 45 min at 37 °C with pipetting every 15 min. Upon washing in PBS, the cell solution containing both epithelial and mesenchymal cells were plated in Matrigel and Basal medium containing ENR added. When epithelial organoids emerged from the mesenchymal co-culture, these were passaged by mechanical disruption approximately once a week. Once stable epithelial organoid cultures were obtained, these were incubated for 24 h in the absence or presence of 10 μM IWR-1-endo (Stemcell Technologies) or 5 μM IWP-2 (Stemgent) or vehicle (DMSO) control starting 4 days after passaging.

**Fetal whole intestine culture**. E13.5 intestines were dissected from the embryos in cold PBS, connective tissue was separated, and intestines were placed on transwells (Costar) in BGJb media (Invitrogen) supplemented with 1% pen/strep (vol/vol) (Invitrogen) and 0.1 mg/mL ascorbic acid. Intestines were cultured for 48 h with or without SAG (1 μM, Enzo Life Sciences) and LGK-974 (1 μM, Selleckchem) at 37 °C with 5% $CO_2$ with media changes every 24 h.

**Gene expression and RNA sequencing**. RNA from purified epithelial and mesenchymal cells, intestinal biopsies or organoids was extracted using the RNeasy Micro kit (Qiagen) according to manufacturers' protocol. For qPCR analyses, cDNA was synthesised using Superscript III reverse transcriptase (Invitrogen) and random primers. qPCR was performed with PowerUp SYBR Green Master Mix (ThermoFischer) using the QuantStudio 6 Flex Real-Time PCR System (Life technologies). CT value was normalised to *Epcam* (intestinal biopsies) or *Gapdh* (epithelial organoid cultures), using the ΔCt method. Primer sequences were as follows: *Epcam* fw: 5-AACACAAGACGACGTGGACA-3, *Epcam* rev: 5-GCTCT CCGTTCACTCTCAGG-3; *Shh* fw: 5-ACTCACCCCCAATTACAACC-3, *Shh* rev: 5-ACAGAGATGGCCAAGGCATTTA-3; *Gapdh* fw: 5-TGTTCCTACCCCCAA TGTGT-3; *Gapdh* rev: 5-TGTGAGGGAGATGCTCAGTG-3; *Lgr5* fw: 5-ACCCG CCAGTCTCCTACATC-3, *Lgr5* rev: 5-GCATCTAGGCGCAGGGATTG-3; *Axin2* fw: 5-TGAAACTGGAGCTGGAAAGC-3, *Axin2* rev: 5-AGAGGTGGTCGTCCAA AATG-3.

For RNA-seq libraries, the quality of the RNA was assessed on an Agilent Technologies Bioanalyzer 6000 RNA pico chip (Agilent) as per the manufacturer's instructions. Up to 10 ng of RNA was used as input material and libraries were prepared according to the SMARTer Stranded Total RNA-Seq kit v2 Pico Input Mammalian kit (Takara). In brief, samples were fragmented at 94 °C for 3 min prior to first-strand synthesis. Illumina adaptors and indexes were added to single-stranded cDNA via 5 cycles of PCR. Libraries were hybridised to R-probes for fragments originating from ribosomal RNA to be cleaved by ZapR. The resulting ribo-depleted library fragments were amplified with 10 cycles of PCR. Fragment length was confirmed with the Bioanalyzer DNA High Sensitivity Kit (Agilent) and concentration was quantified with the Qubit dsDNA HS Assay (ThermoFisher Scientific). Finally, samples were sequenced on the Illumina NextSeq 500 platform (Illumina).

**RNA-seq analyses**. Mapping and quantification of fastq files were done via the Rsubread package (version 2.0)[55]. Reads were mapped to the mm10/GRch38 genome assembly using the align-function with default settings, except nBestLocations = 16 and nTrim5 = 6. Mapped reads quantified across RefSeq gene models using the featureCounts-function with default settings, except strandSpecific = 2. This approach retains multi-mapping reads for gene-level quantification, as long as only one of the multiple locations maps to an annotated gene.

Differentially expressed (DE) gene analysis was carried out using edgeR (version 3.28)[56,57] and limma (version 3.42)[58] with default settings unless otherwise noted. The Wnt-inhibition experiment was analysed separately but included 9 EpCAM⁺ samples from the main experiment. The model matrix was set as intercept-free design by defining groups as cell type-segment combinations (e.g. EpCAM⁺, PDGFRaʰⁱᵍʰ, Proximal, Mid, etc), and including mother as a blocking variable (~0+Group+Replicate). Lowly expressed genes were removed with the filterByExpr (approximately at least 10 counts-per-million when accounting for library size and study design) using this model matrix, after which normalisation factors were estimated using calcNormFactors. The limma-voom pipeline was used with contrasts set as pairwise comparisons between segments within each celltype (treat was run with robust = TRUE). F-tests across multiple contrast were extracted using topTable (eBayes was run with robust = TRUE). In analyses, we employed the following cut-off to identify up- or down-regulated genes: log2 Fold change (FC) > 0.5 or <0.5 and adjusted *p*-value < 0.05.

RPKM-values were extracted using rpkm and rpkmByGroup. Heatmaps were drawn using pheatmap (version 1.0.10) with z-scores calculated from rpkm values. GO analyses was performed via goana function from limma using gene length as covariate. Euler diagrams were generated using the eulerr package (version 6.0.0).

**ChIP-sequencing**. Sorted EpCAM⁺ cells were pelleted and fixed in 1% formaldehyde in PBS for 10 min at room temperature before the addition of Glycine to a final concentration of at least 0.125 M. Fixed chromatin was sheared by adaptive focused acoustics technology using a M220 Focused-ultrasonicator (Covaris) and the truChIP Chromatin Shearing Kit (Covaris) according to manufacturer's instructions until fragments of length 200–500 bp were obtained. Immunoprecipitation was performed with anti-TCF7L2 (C48H11, #2569, Cell Signaling) or rabbit IgG (Abcam) in IP buffer (100 mM NaCl, 0.26% SDS, 1.67% Triton X-100, 75 mM Tris-HCl (pH 8.0), 5 mM EDTA, 0.02% NaN3 and 1% glycerol) overnight rotating at 4 C. Subsequently, a mix of Protein A and Protein G dynabeads (TermoFischer Scientific) blocked with BSA was added and after 3 h incubation, beads were washed 4x with low-salt wash buffer (150 mM NaCl, 1% Triton X-100, 0.1% SDS, 2 mM EDTA, 20 mM Tris-HCl (pH 8.0)), 3x with high-salt wash buffer (500 mM NaCl, 1% Triton X-100, 0.1% SDS, 2 mM EDTA, 20 mM Tris-HCl (pH 8.0)), 1x with LiCl Buffer (250 mM LiCl, 10 mM Tris-HCl, 1 mM EDTA, 0.5% Sodium Deoxycholate, 0.5% NP-40), and 1x in TE (10 mM Tris-HCl, 1 mM EDTA). Beads were incubated at 37 C shaking in TE with 50 ug/ml RNAse (Thermo Fischer Scientific) for 1 h before the addition of SDS to a final concentration of 0.5% and 0.5 mg/ml Proteinase K (Thermo Fischer Scientific). After overnight incubation at 37C, samples were incubated for another 9 h at 65C prior to purification with minElute PCR purification kit (Qiagen). Concentration was estimated using a Nanodrop 3300 Fluorospectrometer (ThermoFischer) and the Picogreen Assay for dsDNA reagent kit (Molecular Probes). Adaptor-ligated libraries were generated using the Accel-NGS 2S Plus DNA Library kit for Illumina (Swift Bioscience) in combination with the 2S Indexing Kit for Illumina (Swift Bioscience) according to manufacturer's instructions, fragment length was confirmed with the Bioanalyzer DNA High Sensitivity Kit and concentration determined with the Qubit dsDNA HS Assay before sequencing on the Illumina NextSeq 500 platform (Illumina).

**ChIP-seq data analyses**. Concatenated reads were trimmed using Trim Galore! (Galaxy version 0.4.2) with trimming of "Illumina Universal" adaptor sequences and standard settings except N = 2 (remove 2 bp from the 3' end), and N = 25 (Discard reads that became shorter than length 25). Trimmed reads were mapped to mm10 canonical (April 2020) using Map with Bowtie for Illumina (Galaxy version 1.2.0) with standard settings except m = 1 (supress all alignments if >1 exist). After SAM-to-BAM conversion, duplicate reads were removed using the RmDup function (Galaxy Version 2.0.1). Peak calling was performed using MACS2 callpeak (Galaxy Version 2.1.1.20160309.6) using the IgG ChIP sample as negative control and standard settings except effective genome size = *M. musculus* (1.87e9). BAM files and MACS2 output files were imported into Easeq (version 1.111)[57], which was used for the generation of plots and annotation of peaks to mm10 Refseq genes using the UCSC table browser retrival tool.

**Statistics**. The number of biological and technical replicates and the number of animals is indicated in figure legends and text. All tested animals were included. The sample size was not predetermined. For all experiments with error bars standard deviation (s.d.) was calculated to indicate the variation within each experiment or sample. Statistical tests were used to assess significance as indicated in individual figure legends.

**Reporting summary**. Further information on research design is available in the Nature Research Reporting Summary linked to this article.

## Data availability

The RNA-seq and ChIP-seq data generated in this study have been deposited in the NCBI's Gene Expression Omnibus (GEO) database under accession code GSE183671. Lists of differentially expressed genes as well as annotated ChIP-seq peaks are provided in Supplementary Data files. The sequence of the MACS1 enhancer used in this study was retrieved from DNA Data Bank of Japan (DDBJ) under accession number AB453049[42]. Source data are provided with this paper.

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

## Acknowledgements

We thank Lydia Sorokin (Institute of Physiological Chemistry and Pathobiochemistry, University of Muenster), Anthony Koleske (Department of Molecular Biophysics and Biochemistry, Yale University), Jeffrey H. Miner (Division of Nephrology, Washington University School of Medicine), Hironobu Fujiwara (R Center for Biosystems Dynamics Research), Brigid Hogan (Duke University Medical Center) Wim de Lau and Hans Clevers (Hubrecht Institute) for providing reagents, the animal caretakers at the MGU and Biocenter, and members of the Jensen lab for comments and technical assistance. This work was supported by European Union's Horizon 2020 research and innovation programme (grant agreement INTENS 668294 and STEMHEALTH ERCCoG682665, KBJ), the Novo Nordisk Foundation (NNF17OC0028730, NNF18OC0034066 and NNF20OC0064376, KBJ), The Danish Cancer Society (R124-A7724), the Marie Curie fellowship programme (656099/H2020-MSCA-IF-2014, JG), the EMBO Long-Term fellowship (ALTE 946-2019, RBB), the Howard Hughes Medical Institute (HHMI), Mathers Foundation (KCG), Ludwig Foundation (KCG), and NIH (1R01DK115728-01A1, KCG). The Novo Nordisk Foundation Center for Stem Cell Biology and Novo Nordisk Foundation Center for Stem Cell Medicine are supported by Novo Nordisk Foundation grants (NNF17CC0027852 and NNF21CC0073729).

## Author contributions

M.M., M.T.P. and K.B.J. designed the experiments. M.M., M.T.P. performed the experiments with assistance from J.G., R.B.B., Y.A. and P.J.S. M.M., M.T.P., J.D., L.R. and M.T. analysed the data. P.S. provided the Top.CFP mouse. M.M., M.T.P. and K.B.J. interpreted the results. Y.M., C.G., A.S. and P.S. provided resources. M.M., M.T.P. and K.B.J. wrote the manuscript with contributions from all authors.

## Competing interests

KCG has licensed IP related to surrogate Wnt agonists to Surrozen Therapeutics. All other authors declare no competing interests.
