## [Peer Review File · Nature Communications]

Mesenchymal-epithelial crosstalk shapes intestinal regionalisation via Wnt and Shh signallingREVIEWER COMMENTS

Reviewer #1 (Remarks to the Author):

This study employs transcriptome profiling, CHIPseq analysis and organoid culture to investigate the underlying mechanism directing regional specification in the developing mouse small intestine. Evidence is presented for an interplay between canonical Wnt signaling and Hedgehog signaling in the developing epithelium and underlying mesenchyme being critical in directing anterior-posterior regionalization of the SI.

The study makes some interesting observations, which would be potentially of interest to the field, but lacks clarity in some key areas.

Specific critique:

- 1) Intestinal development is known to proceed in an anterior to posterior wave, with the duodenum maturing into the crypt-villus structures earlier than the ileum. I then wonder whether the transcriptomic comparisons made for the different regions at E16.5 are simply reflecting this temporal delay in re-organization of the nascent epithelium? If the proximal SI epithelium at E12.5 was compared to its E12.5 distal epithelium, would the observed changes in Wnt signaling, HH signaling etc be so apparent?
- 2) As a general remark, there is an over-reliance on q-PCR data to highlight the differences in gene expression observed, which may fail to reveal more subtle changes in localization of the target genes being studied. For some of the key findings, eg, Shh1 expression changes, I think either IHC or in-situ should be included to underscore the key findings.
- 3) Figure 1- It is difficult to appreciate the relative location of the PDGFR^{high} and med populations in the developing SI from the supplied IF images. Can higher resolution images be supplied that highlight this? What was the rationale for comparing the different populations based on their PDGFR^a expression levels?
- 4) Figure 1f – Do the PDGFR^{med} cells also support the organoid growth in the absence of ENR? This would support a functional difference in addition to a transcriptomic one compared with their PDGFR^{high} counterparts. Can the PDGFR^{med} cells also support growth of mid and distal SI organoids in the absence of ENR? Conversely, are the PDGFR^{high} cells isolated from different regions of the SI equally proficient at supporting ENR-free organoid growth? Can mesenchymal cells from different regions alter regional specification of the co-cultured organoids? Can these organoids be passaged, or is it a transient effect? Does addition of PORCNI to the PDGFR^{high}-supplemented cultures reverse the growth promoting effect? Given that the control organoids look different to the organoids grown in the

presence of PDGFR^{high} cells, it would be useful to better characterize the organoids in terms of differentiation potential, proliferation activity etc.

5) Fig2 – please validate the key Wnt effector molecules by RNAscope/IF. Are there no differences in Fzd expression in the different regions?

6) Fig 2 – Given the TCF-regulated CFP expression in the adult SI appears to be restricted to the CBC stem cells, how confident are the authors that this is sensitive enough to report all Wnt-active cells? The TA compartment should also be lighting up here.

7) Fig 2c – The described difference in CFP expression in the distal versus mid intestine is not readily visible – it appears broader within the epithelium, but not appreciably higher per cell. Can this be properly quantified to substantiate the claims? Given the potential sensitivity issue of this reporter, it would be helpful to independently document the level of Wnt activity by performing nuclear beta-catenin IHC on the different regions.

8) Fig 2d – do the organoids demonstrate region-specific differences in their sensitivity to Wnt inhibitors ex vivo? Why do there appear to be fewer organoids forming from the proximal SI when the ENR medium is supplemented with Wnt surrogates? Can these organoids be passaged long term?

9) Fig 3 – The described lack of effect of a 4 day PORCNI treatment on embryonic/SI development is surprising given the known roles for Wnt signaling during development. Can the authors properly substantiate this, to preclude the contribution of systemic effects of blocking Wnt signaling on SI development. Please show TOP-CFP expression on high quality sections from each region of control and treated developing SI.

10) Fig 3b – were other signaling pathways than just Wnt represented amongst the differentially regulated genes following PORCNI treatment? This would give a better indication as to the potential (indirect) contribution of other pathways following the extended Wnt signaling blockade.

11) Fig 3 – better characterization of the SI phenotype following PORCNI treatment is needed to substantiate the conclusions. What was the effect on the relative lengths of the different regions? More pronounced in distal versus proximal? What was the effect on proliferation, differentiation etc. Does treating pregnant mice with PORCNI at earlier time-points (say E10.5) magnify the effects on the proximal SI, which matures first?

12) Fig 4 – Whilst TCF4 is the major TCF involved in driving Wnt signaling during SI development, it is not the only one. Would using a pan TCF antibody therefore not be expected to reveal a more comprehensive set of Wnt target genes for evaluation?

13) Fig 4a – Why was a different PORCN inhibitor used here? I am surprised that treatment with Wnt inhibitors did not affect organoid viability – can this be better substantiated? If the CHIP/transcriptomic data are correct and shh is a direct Wnt target gene, it should be rapidly (<12 hours) switched off in E16.5 distal SI organoids treated with the PORCNI/tankyrase inhibitor. Can this be tested? It would also be interesting to see whether the Wnt inhibitor experiments using organoids deliver region-specific results (ie, does distal SI organoid respond differently to prox SI organoids)? Is Shh lost from the secretome of distal SI organoids cultured with the PDGFR^{high} mesenchymal cells in the presence of the Wnt inhibitors?

--

Reviewer #2 (Remarks to the Author):

In this manuscript, the authors demonstrated that the subepithelial mesenchymal cells efficiently build the structure of the developing small intestine by regulating Wnt signaling activity. Additionally, the authors identified that PDGFR^{high} cells contribute to maintaining the relatively higher activity of Wnt signaling at the distal region of the small intestine, which in turn regulates Shh on the mesenchymal cells driving the villus formation. Despite some convincing results, the overall experimental approaches are descriptive. The crucial roles of mesenchymal cells in maintaining Wnt signaling in the intestine were well documented. The crosstalk between Wnt and Shh signaling in tissue homeostasis and regeneration was previously reported.

Major Points

- PDGFR^{high} cells were selected as cells secreting Wnt agonists and are responsible for the maintenance of relatively higher Wnt signaling activity in the distal region. Beyond this descriptive approach, additional *in vivo* evidence should be provided to claim the importance of PDGFR^{high} cells in Wnt hyperactivation.
- The reason for not choosing PDGFR^{Med} cells (due to Lrig1 expression) should be clearly explained.
- Figure 3b. Upon PORCNI treatment, the authors showed Axin2 change. What about the other Wnt/ β -catenin target genes?
- Line 173-179 and Figure 4e, f. The authors showed that "Shh expression is directly regulated by WNT signaling in the developing distal SI." To prove this, the authors should at least perform an experiment like figure 3d, upon PORCNI treatment; can Shh overexpression recover the villus formation?

Minor Points

- Grammatical integrity can be improved. A typo (growt).
- Please add the details on organoid culture protocol.
- Missing scale bars in the images (e.g., organoids)

--

Reviewer #3 (Remarks to the Author):

In this manuscript the authors interrogate the mesenchymal population of developing mouse small intestine and identify a cell population, expressing high amounts of PDGf receptor, governing specific mesenchymal gradient expression of Wnt, being responsible for proximal-distal regionalization of developing mouse small intestine.

Overall the paper is nicely written and easy to read and follow. My points below represents questions about experimental design and concerns about data interpretation

Questions/Concerns

1. In figure 1F authors present evidence to encourage the conclusion of PDGFR^{high} population supporting the fetal organoid growth. It would be fair to include in this comparison the PDGFR populations from proximal, middle and distal segments. It would be helpful to include PDGFR medium population as well in order to substantiate the authors claims that PDGF high population is functionally very different to PDGF low population.
2. Authors use sequencing data to extrapolate the expression patterns of various genes throughout the manuscript. These data are stated to be generated from 3 biological replicate which would mean 3 embryos probably coming from the same litter. If this is indeed the case, I wonder how representative this data is? Would the same expression patterns hold on in various litters? In my opinion the data (at least a part shaping the main conclusions) would gain more strength if conformed in the second litter by qPCR analyses.
3. I wonder whether the picture represented in figure 2C for the proximal intestinal segment is representative? The percentage of sorted cells (extended data Fig. 2b) is around 40 % while figure 2C proximal hardly has any signal from Top. CFP
4. I was puzzled by the data represented in figure 2D-F where authors state that organoids from middle and distal 16.5 epithelial cells can not be cultured in ENR medium without supplementing it with surrogate Wnt. Previous data from the same group (Forham et al, 2013) states that organoids from similar stage E16 can be cultured in the absence of Wnt and even in the presence of wnt inhibitor DKK1. In that paper the authors conclude: "FenS can be maintained independently of Wnt signaling. This distinguished them from adult organoids". Could authors comment on this and explain how these two studies align with each other with respect of culture conditions?
5. On the same page with concern 4, in the current manuscript, authors use distal organoid cultures to assess the effect of Wnt signaling on SHH expression in distal organoid cultures in figure 4f. They utilize two different wnt inhibitors and claim no effect on organoid growth/morphology albeit the Shh reduction. The micrographs of organoid cultures were not shown. Can the authors comment how do these data relate with conclusions of figure 2d-f?
6. Authors use the porcni inhibitor of Wnt secretion to assess whether inhibition of Wnt signaling in vivo has region specific differences for small intestinal development and find the most profound effect for the distal intestine. However by inspection of extended data Figure 2b, one can deduce that their treatment worked the best in the distal intestine. The treatment in the proximal intestine was the least

efficient, less the 50%, while in the distal intestine the treatment worked at 90% efficiency. How can the authors be sure that the difference they observe is simply not due to the different efficiency of the treatment in three intestinal segments. This particularly concerns me as the inhibition of villi formation was evident in the proximal and middle part and the effect correlates with the efficiency of the treatment.

7. Furthermore, wnt signaling is historically described to control the proliferation/differentiation of the intestinal epithelial cells (Korinek et al, 1998). Therefore porcni treatment can cause massive loss of proliferatin intestinal epithelial cells in intervillus pockets, and consequentially blunting of the villi. The authors should look at proliferation/differentiation upon porcni treatment. In other words there could be other reasons for the phenotype authors observe upon using porcni

Reviewer #1:

This study employs transcriptome profiling, CHIPseq analysis and organoid culture to investigate the underlying mechanism directing regional specification in the developing mouse small intestine. Evidence is presented for an interplay between canonical Wnt signaling and Hedgehog signaling in the developing epithelium and underlying mesenchyme being critical in directing anterior-posterior regionalization of the SI.

The study makes some interesting observations, which would be potentially of interest to the field, but lacks clarity in some key areas.

We thank the reviewer for her/his suggestions and that she/he finds our study interesting and of interest to the field. Addressing the comments has significantly enhanced the clarity of the manuscript.

Specific critique:

1) Intestinal development is known to proceed in an anterior to posterior wave, with the duodenum maturing into the crypt-villus structures earlier than the Ileum. I then wonder whether the transcriptomic comparisons made for the different regions at E16.5 are simply reflecting this temporal delay in re-organization of the nascent epithelium? If the proximal SI epithelium at E12.5 was compared to its E12.5 distal epithelium, would the observed changes in Wnt signaling, HH signaling etc be so apparent?

We have performed RNAseq analysis of the proximal and distal epithelium from E13.5, reflecting the very last stage before villus formation. Here we observe that Shh, Axin2 and Lgr5 are already expressed in a gradient along the rostro-caudal axis similar to what can be observed at E16.5. This strongly suggests that the Wnt signalling gradient precedes morphogenesis and is not simply reflecting a temporal delay. This is now included in Suppl. Figure 7e-f.

2) As a general remark, there is an over-reliance on q-PCR data to highlight the differences in gene expression observed, which may fail to reveal more subtle changes in localization

of the target genes being studied. For some of the key findings, eg, Shh1 expression changes, I think either IHC or in-situ should be included to underscore the key findings.

We have now performed immunohistochemistry for Shh and observe that it is indeed detected at higher levels in the distal epithelium. This is included as Figure 4c. We have also included in situ hybridisation for WNT signalling components. This is included as Suppl. Fig. 3b, d.

3) Figure 1- It is difficult to appreciate the relative location of the PDGFR^{high} and med populations in the developing SI from the supplied IF images. Can higher resolution images be supplied that highlight this? What was the rationale for comparing the different populations based on their PDGFR^a expression levels?

We are grateful for the suggestion and have now included images at higher magnification. We also now explicitly explain the rationale for why we have chosen this population for our deeper analysis. This is included in Figure 1a and Suppl. Fig 1a, and we have now included a description in the text why these markers have been chosen.

4) Figure 1f – Do the PDGFR^{med} cells also support the organoid growth in the absence of ENR?

We provide evidence demonstrating that indeed the PDGFR^{med} also support organoid growth, however, not to the same degree as the PDGFR^{high} population. This is included as Figure 1f-g. This indeed supports that the populations are not only transcriptionally different but also functionally different.

This would support a functional difference in addition to a transcriptomic one compared with their PDGFR^{high} counterparts. Can the PDGFR^{med} cells also support growth of mid and distal SI organoids in the absence of ENR?

The PDGFR^{med} population does not support the growth of organoids from the mid and distal region of the intestine.

Conversely, are the PDGFR^{high} cells isolated from different regions of the SI equally proficient at supporting ENR-free organoid growth?

The PDGFRa^{high} cells isolated from the different regions are equally proficient in supporting the growth of organoids derived from the proximal part of the small intestine.

Panel a,b: Purified EpCAM⁺ cells from proximal SI were co-cultured with PDGFRa^{high} cells isolated from proximal, mid and distal regions. Subsequently, the emergence of fetal organoids was quantified.

Can mesenchymal cells from different regions alter regional specification of the co-cultured organoids?

This is an interesting question, however, extremely difficult to perform. We isolate cells from the different regions from one litter of embryos (7-8 embryos) by flow cytometry. This provides enough cells for 1 well for the PDGFRa^{high} cell population from each segment. This droplet of matrigel contains both mesenchymal and epithelial cells. Given that we have no antibodies for markers associated with the different segments, we would need to address this question by repurifying epithelial cells from a single well and perform either sequencing or qPCR. This is currently not feasible.

Can these organoids be passaged, or is it a transient effect?

All our organoid lines can be passaged when supplemented with EGF, Noggin and R-spondin in subsequent passages. The culture conditions do not support propagation of mesenchymal cells, and hence the effect of the fibroblasts is transient.

Does addition of PORCNI to the PDGFRa^{high}-supplemented cultures reverse the growth promoting effect?

We performed this experiment, and we observe that PDGFRa^{high}-supplemented cultures isolated from the proximal half of the SI can be propagated in the presence of PORCNI. This

is aligned with our previous observations that the proximal intestinal epithelial cells are less dependent on Wnt stimulation for growth when compared to epithelial cells from the mid/distal segments (Fordham et al., 2013, Cell Stem Cell; also see 1st submission).

Panel c,d: Purified EpCAM⁺ and PDGFRa^{high} cells were co-cultured with or without PORCNI. Subsequently, the growth of fetal organoids was noted.

Given that the control organoids look different to the organoids grown in the presence of PDGFRa^{high} cells, it would be useful to better characterize the organoids in terms of differentiation potential, proliferation activity etc.

We thank the reviewer for this comment. We do, however, utilise this experiment as a functional assay for testing the properties of the PDGFRa^{high} population. It is clear from these experiments that these mesenchymal cells support growth of epithelial cells and based on the data provided in the revised version of the manuscript also better than the PDGFRa^{med} population. We never claim that these organoids are identical to organoids grown under EGF/Noggin/Rspondin conditions, as the mesenchymal cells are likely to express additional growth factors at different levels. We consequently feel that characterising the growing epithelial cell population does not add significant value to the conclusions that we draw from this experiment.

5) Fig2 – please validate the key Wnt effector molecules by RNAscope/IF. Are there no differences in Fzd expression in the different regions?

We have included in situ hybridisation for WNT effector molecules. This is included as Suppl. Fig. 3b, d. We also show RNA-seq data for the Fzd receptors in Suppl. Fig. 3e. Only Fzd10 is significantly differentially expressed.

6) Fig 2 – Given the TCF-regulated CFP expression in the adult SI appears to be restricted to the CBC stem cells, how confident are the authors that this is sensitive enough to report all Wnt-active cells? The TA compartment should also be lighting up here.

This is obviously a consideration for any reporter used for gene activation. This is based on our experience a very sensitive reporter strain, and it has been used by others for studies of Wnt signalling in intestinal epithelial cells showing excellent alignment with other Wnt target genes (Basak et al., 2017, Cell Stem Cell). We have performed a careful analysis of the adult intestinal crypts, where we analysed numerous tissue samples and observed that the levels of CFP from the reporter is characterised by a gradient showing highest expression at the bottom of the crypt extending into the TA compartment as expected. We have updated the Suppl. Fig 4c.

7) Fig 2c – The described difference in CFP expression in the distal versus mid intestine is not readily visible – it appears broader within the epithelium, but not appreciably higher per cell. Can this be properly quantified to substantiate the claims?

This is what we observe, and we apologise for any confusion. We have now tried to phrase this in a more concise manner in the revised manuscript. Our flow cytometry data show that the intensities are very similar between epithelial cells isolated from the different segments, but that the proportion of positive cells is much higher in the distal part of the small intestine. This is quantified in the flow cytometry data. We now provide images at higher resolution to support this (Figure 2c).

Given the potential sensitivity issue of this reporter, it would be helpful to independently document the level of Wnt activity by performing nuclear beta-catenin IHC on the different regions.

The validity of the reporter data is supported by the expression analysis and the inhibitor experiments. The activity of the Wnt reporter is aligned with our previous characterisation of Lgr5 expression in the developing intestinal epithelium using two reporter models (Lgr5-eGFP-iDTR; Lgr5-eGFP-ires-CreERT2), where there is much more prominent expression of the Wnt target gene Lgr5 in the distal epithelium (Guiu et al., 2019, Nature). Similarly, high levels of WNT signalling was previously reported by the group of Jason Spence in the distal developing epithelium (Chin et al., 2016, Stem Cell Reports). Thus, we believe this

serves as independent documentation of the differences in Wnt signalling along the rostral-caudal axis of the intestine. We appreciate that beta-catenin translocates to the nucleus upon Wnt activation, however, staining for beta-catenin and quantification of its translocation are currently not straight-forward. This can be exemplified by a recent study, where beta-catenin in homeostatic tissue is detected in the nucleus of Paneth cells. These cells have been described as a source of Wnt in the epithelium and not cells having active Wnt signalling. In contrast, in the same manuscript high levels of nuclear beta-catenin is detected in cells lacking the tumour suppressor APC (Flanagan et al., 2021, Nature). Moreover, nuclear beta-catenin does not necessarily imply transcriptional activation (See Maretto et al., 2003, PNAS).

8) Fig 2d – do the organoids demonstrate region-specific differences in their sensitivity to Wnt inhibitors ex vivo? Why do there appear to be fewer organoids forming from the proximal SI when the ENR medium is supplemented with Wnt surrogates?

The growth of organoids from all parts of the intestine are affected by the porcupine inhibitor if cells are plated in PORCNI directly upon isolation. Indeed, there appear to be slight differences, but these are not statistically significant and we would refrain from speculations (Figure 2f). This is however aligned with our study showing greater effects of Wnt on epithelial cells from the distal part.

Can these organoids be passaged long term?

All our established organoids can be propagated long-term.

9) Fig 3 – The described lack of effect of a 4 day PORCNI treatment on embryonic/SI development is surprising given the known roles for Wnt signaling during development. Can the authors properly substantiate this, to preclude the contribution of systemic effects of blocking Wnt signaling on SI development. Please show TOP-CFP expression on high quality sections from each region of control and treated developing SI.

We apologise for the confusion and we have now emphasised these aspects in the revised manuscript. There are substantial defects in the formed embryos upon PORCNI, defects that are aligned with tissues where we know from previous studies that Wnt activation plays a role. This includes digit formation, eye-lid closure, hair and mammary gland formation, as

well as small intestinal defects. This is included in Figure 3. We quantify TOP-CFP expression using flow cytometry, which to our knowledge is the most unbiased and quantitative method for this type of analysis. Obviously, we cannot exclude systemic effects, however, our ex vivo and organoid data strongly support the in vivo data, substantiating our conclusions (Figure 4). It is also worth noting that prior studies have administered porcupine inhibitor for up to 30 days without seeing pronounced changes (Huels et al., 2018, Nature Communications).

10) Fig 3b – were other signaling pathways than just Wnt represented amongst the differentially regulated genes following PORCNI treatment? This would give a better indication as to the potential (indirect) contribution of other pathways following the extended Wnt signaling blockade.

As expected, we do see secondary impact on other signaling pathways. This includes down-regulation of Id1 and Id3 suggesting impaired BMP signaling (Suppl. Data 7j) which is in agreement with the literature reporting link between Bmp ligands and Shh (Shyer et al., 2015, Cell).

11) Fig 3 – better characterization of the SI phenotype following PORCNI treatment is needed to substantiate the conclusions. What was the effect on the relative lengths of the different regions? More pronounced in distal versus proximal?

We indeed thank the reviewer for this question, however, this is a difficult question to address. We have not been able to divide the fetal intestine convincingly into duodenum, jejunum and ileum, hence we work with material from defined spatial regions as outlined in our materials and methods section (proximal, mid and distal). In order to address whether the intestine grows in an anisotropical manner along its length we would have to perform quantitative fate mapping studies looking at clone size variation over time. We have previously performed such studies and estimate that this will take more than 2 years to complete this in an appropriate quantitative manner, which we feel is beyond the scope of this study.

What was the effect on proliferation, differentiation etc.

Analysis of material from tissues from PORCNI treated animals show that the number of proliferating cells is reduced, but proliferation is clearly not abrogated. We see this effect in both the proximal and distal part of the intestine reinforcing that the inhibitor is active in the entire small intestine (Suppl. Data 5f). Given that the intestinal epithelium is composed of equipotent progenitors and that the cells normally do not express terminal differentiation markers at this late stage of development we cannot access the differentiation status (Guiu et al., 2019, Nature).

Does treating pregnant mice with PORCNI at earlier time-points (say E10.5) magnify the effects on the proximal SI, which matures first?

We show that WNT signalling is represented by a gradient in the epithelium, which we now show is already visible at E13.5 by RNAseq analysis of the epithelium. Additionally, our in vitro organoid studies support the conclusion that Wnt signalling plays a stronger role in the distal SI. Performing a PORCNI experiment starting at E10.5 will not change the major conclusion of our study, as WNT signalling will still be essential for patterning of the distal small intestine from E12.5 and onwards. Therefore, we have not performed this in vivo experiment but instead focused on the ex vivo rescue experiments, where we see pronounced effects on villus formation upon PORCNI treatment, which can be rescued by Smoothed agonists (SAG).

12) Fig 4 – Whilst TCF4 is the major TCF involved in driving Wnt signaling during SI development, it is not the only one. Would using a pan TCF antibody therefore not be expected to reveal a more comprehensive set of Wnt target genes for evaluation?

The knockout data clearly demonstrate that TCF4 is the major effector of beta-catenin in the intestinal epithelium (Korinek et al., 1998, Nat Genetics), and although it will provide a more comprehensive view on potential binding sites, it will not add to the seminal discovery that Shh is a target of WNT activation.

13) Fig 4a – Why was a different PORCNI inhibitor used here? I am surprised that treatment with Wnt inhibitors did not affect organoid viability – can this be better substantiated? If the CHIP/transcriptomic data are correct and shh is a direct Wnt target gene, it should be rapidly

(<12 hours) switched off in E16.5 distal SI organoids treated with the PORCNI/tankyrase inhibitor. Can this be tested?

In order to address this question, we have treated intestinal epithelial organoids with both porcupine and tankyrase inhibitors to show that this is an effect of beta-catenin directed Wnt mediated signalling. We show qPCR data for the 24h time point to avoid confounding effects related to e.g., transcript stability (Fig 4f and Suppl. Data 7g). Within this short time frame, we do not see effects on organoid viability.

It would also be interesting to see whether the Wnt inhibitor experiments using organoids deliver region-specific results (ie, does distal SI organoid respond differently to prox SI organoids)?

We agree and we have now performed inhibitor experiments with organoids derived from the proximal, mid and distal part of the small intestine. We observe the prominent effects on the distal part where the level of activation is also higher (Suppl. Data 7g).

Is Shh lost from the secretome of distal SI organoids cultured with the PDGRA^{high} mesenchymal cells in the presence of the Wnt inhibitors?

This would indeed be an interesting experiment, however, technically very difficult if not impossible.

1) The limitations in the number of cells we can collect and grow from a single litter makes it very difficult to generate enough material for this type of analysis.

2) Organoids from the distal small intestine require Wnt signalling to grow. If we inhibit the pathway from seeding the cells, we will not be able to grow any cells. On the other hand if we culture the cells for a given period of time and then add the inhibitor, we would only be able to assess the effects from the addition of the inhibitor and then until the time of analysis under conditions where the cells have stopped proliferating. The Shh secreted before the addition of the inhibitor would consequently influence the outcome significantly.

Reviewer #2:

In this manuscript, the authors demonstrated that the subepithelial mesenchymal cells efficiently build the structure of the developing small intestine by regulating Wnt signaling activity. Additionally, the authors identified that PDGFR^{high} cells contribute to maintaining the relatively higher activity of Wnt signaling at the distal region of the small intestine, which in turn regulates Shh on the mesenchymal cells driving the villus formation. Despite some convincing results, the overall experimental approaches are descriptive. The crucial roles of mesenchymal cells in maintaining Wnt signaling in the intestine were well documented. The crosstalk between Wnt and Shh signaling in tissue homeostasis and regeneration was previously reported.

We thank the reviewer for her/his excellent comments and constructive criticism and that reviewer finds our data convincing and well documented. We feel that addressing these comments has significantly improved the manuscript.

The reviewer points out that Wnt and Shh signalling have previously been linked during homeostasis and regeneration e.g. hair follicle formation and regrowth. However, to our knowledge it has never been reported that Tcf7L2, the key transcriptional regulator bound by beta-catenin, binds to the MACS1 enhancer, nor that a crosstalk between Wnt signals from mesenchymal cells promotes Shh expression in the epithelium. We strongly believe that the direct link between Wnt activation and Shh expression in the epithelium will have widespread implications on our understanding of developmental processes, homeostasis and regeneration.

Major Points

- PDGFR^{high} cells were selected as cells secreting Wnt agonists and are responsible for the maintenance of relatively higher Wnt signaling activity in the distal region. Beyond this descriptive approach, additional in vivo evidence should be provided to claim the importance of PDGFR^{high} cells in Wnt hyperactivation.

We thank the reviewer for this comment. In an attempt to specifically target this population of cells in vivo we have imported a PDGFRaCreER^{T2} mouse model (Chung et al., 2018,

Development). When labelling cells at E14.5 and E15.5 using PDGFRaCreER^{T2} mice crossed with the ROSA26^{mTmG} model we find that new tools will have to be developed in order to address this question and it is unclear even with such tool whether this question can be address in vivo.

1. Using this mouse model, we cannot exclusively label the subepithelial mesenchyme (see below) as the rest of the mesenchyme is also labelled to some degree.
2. We see that proliferation (as read-out by 1h EdU pulse prior to euthanization of animals) occurs in the entire mesenchyme and there is currently no evidence to support that the subepithelial mesenchyme represents a self-maintained population. It is consequently very likely that during this late stage of development with ongoing morphogenesis that there is a constant exchange of cells between compartments.

Panel e: Labelling of PDGFRa-cells was induced at E14.5 or E15.5 in PDGFRaCreER^{T2};ROSA^{mTmG} animals. 1 hour prior to euthanization the animals were pulsed with EdU.

However, we have now functionally characterised PDGFRa^{high} and PDGFRa^{med} subpopulations isolated directly from the tissue in terms of their ability to support the growth of epithelial cells. Here we show that PDGFRa^{high} cells are superior in their growth promoting ability (Fig. 1f-g).

- The reason for not choosing PDGFRaMed cells (due to Lrig1 expression) should be clearly explained.

We have now performed RNA.seq of PDGFRa^{med}/CD29^{high} population which shows strong overrepresentation of genes associated with muscle function and elevated levels of muscle marker genes Acta2, Myl4 and Des (Suppl Fig 1b-c).

- Figure 3b. Upon PORCNI treatment, the authors showed Axin2 change. What about the other Wnt/ β -catenin target genes?

We have now added the analysis of additional genes Wnt target genes in the supplemental data (Suppl. Fig. 5c). Importantly, these show the same pattern as Axin2.

- Line 173-179 and Figure 4e, f. The authors showed that “Shh expression is directly regulated by WNT signaling in the developing distal SI.” To prove this, the authors should at least perform an experiment like figure 3d, upon PORCNI treatment; can Shh overexpression recover the villus formation?

We thank the reviewer for this excellent suggestion. In order to address this question, we have optimised ex vivo culture conditions for the developing small intestine (Walton et al., 2012, PNAS). Here, we can monitor the process of villi formation under controlled conditions (see also included video). To visualise villi formation we have isolated the whole gut tube from PDGFRaCreER^{T2};ROSA^{mTmG} animals and cultured these in the presence of 4-hydroxy tamoxifen to continuously label the mesenchyme with GFP. In control cultures we observe the emergence of mesenchymal clusters as described previously (Walton et al., 2012, PNAS). Addition of Porcupine inhibitor strongly suppresses cluster formation. Supplementing the cultures with a Sonic hedgehog signalling agonist (SAG) has limited effects on the density of clusters. Importantly, the combination of SAG and PORCNI rescues the effect on cluster formation demonstrating that the direct regulation of Shh by Wnt signalling in the epithelium is essential for villi formation to proceed. This exciting data is now added to the manuscript (Figure 4g-h).

Minor Points

- Grammatical integrity can be improved. A typo (growt).

We apologise for the typos

- Please add the details on organoid culture protocol.

This has now been included

- Missing scale bars in the images (e.g., organoids)

We apologise for the omission

Reviewer #3:

In this manuscript the authors interrogate the mesenchymal population of developing mouse small intestine and identify a cell population, expressing high amounts of PDGf receptor, governing specific mesenchymal gradient expression of Wnt, being responsible for proximal-distal regionalization of developing mouse small intestine.

Overall the paper is nicely written and easy to read and follow. My points below represents questions about experimental design and concerns about data interpretation

We thank the reviewer for the positive and encouraging comments.

Questions/Concerns

1. In figure 1F authors present evidence to encourage the conclusion of PDGFR^{high} population supporting the fetal organoid growth. It would be fair to include in this comparison the PDGFR populations from proximal, middle and distal segments. It would be helpful to include PDGFR medium population as well in order to substantiate the authors claims that PDGF high population is functionally very different to PDGF low population.

We thank the reviewer for this suggestion. Our initial co-culture experiments were performed with cells from the proximal part of the intestine. While we observe that PDGFR^{high} populations from all intestinal regions can support the growth of epithelial cells from the proximal intestine (panels a, b, this response letter), PDGFR^{high} cells sorted from the mid and distal SI are not sufficient to support expansion of cognate epithelial cells in basal medium (panels f,g).

However, supporting our hypothesis that especially the distal PDGFR α^{high} population promote an important Wnt gradient through secretion of RSPONDINS, we find that the addition of Wnt surrogate – in the absence of exogenous EGF, Noggin or RSPONDIN - rescues organoid formation supported by the PDGFR α^{high} population especially for the distal intestine (panels h,i).

While we believe these data strongly support our conclusions, we have chosen not to include the region-specific co-culture experiments in the manuscript, as we find that the results observed for Prox EpCAM+ cells (induced proliferation by Wnt surrogate in the absence of RSPONDINS) suggest that extensive titration of Wnt surrogate would be required to mimic physiological responses.

We have included functional characterization of PDGFR α^{med} population in the manuscript (Figure 1f-g).

2. Authors use sequencing data to extrapolate the expression patterns of various genes throughout the manuscript. These data are stated to be generated from 3 biological replicate which would mean 3 embryos probably coming from the same litter. If this is indeed the case, I wonder how representative this data is? Would the same expression patterns hold on in various litters? In my opinion the data (at least a part shaping the main conclusions) would gain more strength if conformed in the second litter by qpcr analyses.

We completely agree with the reviewer that this is an important point. We have performed all of our analysis on three separate litters to minimise potential variation in developmental stages.

3. I wonder whether the picture represented in figure 2C for the proximal intestinal segment is representative? The percentage of sorted cells (extended data Fig. 2b) is around 40 % while figure 2C proximal hardly has any signal from Top. CFP

This is an excellent point and we have now made this clear in the manuscript that we observe a higher fraction of WNT active cells in the distal part of the small intestine and not cells that are more active. We have also included higher magnification images to visualise this appropriately.

4. I was puzzled by the data represented in figure 2D-F where authors state that organoids from middle and distal 16.5 epithelial cells can not be cultured in ENR medium without supplementing it with surrogate Wnt. Previous data from the same group (Forham et al, 2013) states that organoids from similar stage E16 can be cultured in the absence of Wnt and even in the presence of wnt inhibitor DKK1. In that paper the authors conclude: “FEnS can be maintained independently of Wnt signaling. This distinguished them from adult organoids”. Could authors comment on this and explain how these two studies align with each other with respect of culture conditions?

FEnS from the Fordham study were always derived from the proximal part of the small intestine aligned with our observations here. Indeed, in experiments included in the 2013 manuscript, we also derived organoids from the mid and distal small intestine at postnatal day 2. In line with the observation here, these formed budding organoids, but at a very low efficiency. Based on the current study, this is due to the WNT dependency of intestinal epithelial cells from the distal part of the small intestine, when analysed at E16.5. In addition, the reported experiments from 2013 did not include a sorting step to specifically purify epithelium; we consequently cannot rule out the possibility that upon seeding mesenchymal cells were present in the cultures. These mesenchymal cells are subsequently lost and indeed established FEnS-cultures can be cultured in the absence of WNT pathway activation.

5. On the same page with concern 4, in the current manuscript, authors use distal organoid cultures to assess the effect of Wnt signaling on SHH expression in distal organoid cultures in figure 4f. They utilize two different wnt inhibitors and claim no effect on organoid

growth/morphology albeit the Shh reduction. The micrographs of organoid cultures were not shown. Can the authors comment how do these data relate with conclusions of figure 2d-f? The experiments were performed over the course of 24 hours and not long-term. Indeed, organoids are dependent on WNT signalling and if treated long-term will not be able to grow. We apologise for the confusion and have now made this clear in the text.

6. Authors use the porcni inhibitor of Wnt secretion to assess whether inhibition of Wnt signaling in vivo has region specific differences for small intestinal development and find the most profound effect for the distal intestine. However by inspection of extended data Figure 2b, one can deduce that their treatment worked the best in the distal intestine. The treatment in the proximal intestine was the least efficient, less the 50%, while in the distal intestine the treatment worked at 90% efficiency. How can the authors be sure that the difference they observe is simply not due to the different efficiency of the treatment in three intestinal segments. This particularly concerns me as the inhibition of villi formation was evident in the proximal and middle part and the effect correlates with the efficiency of the treatment.

We apologise for the confusion and have now tried our utmost to clarify this in the resubmitted manuscript.

1. The observation that WNT signalling has prominent functions in the distal small intestine is aligned with prior studies (Guiu et al., 2019, Nature; Chin et al., 2016, Stem Cell Report; Nigmatullina et al., 2017, EMBO).
2. Marker expression of key Wnt target genes in the intestine (Lgr5 and Axin2) are reduced in all segments of the developing small intestine to the same base level.
3. We provide functional evidence that Shh is a Wnt target gene and that a hedgehog signalling agonist can rescue the effect of PORCNI on mesenchymal cluster formation. Given that Shh is expressed prominently in the distal part of the small intestine in a WNT dependent manner this firmly supports our conclusion.
4. Also see point 7

7. Furthermore, wnt signaling is historically described to control the proliferation/differentiation of the intestinal epithelial cells (Korinek et al, 1998). Therefore porcni treatment can cause massive loss of proliferating intestinal epithelial cells in intervillus pockets, and consequentially blunting of the villi. The authors should look at

proliferation/differentiation upon porcni treatment. In other words there could be other reasons for the phenotype authors observe upon using porcni

Analysis of material from tissues from PORCNI treated animals show that the number of proliferating cells is reduced, but proliferation is clearly not abrogated. We see this effect in both the proximal and distal part of the intestine reinforcing that the inhibitor is active in the entire small intestine (Suppl. Data 5f).

REVIEWERS' COMMENTS

Reviewer #1 (Remarks to the Author):

Authors have satisfactorily addressed most of the major critique

Reviewer #2 (Remarks to the Author):

All comments were addressed.

Reviewer #3 (Remarks to the Author):

The authors have answered/clarified my questions/concerns and added new experiments which resulted in significantly improved manuscript

Response to reviewer comments

We are extremely happy with the positive responses from all three reviewers

Reviewer #1 (Remarks to the Author):

Authors have satisfactorily addressed most of the major critique

Reviewer #2 (Remarks to the Author):

All comments were addressed.

Reviewer #3 (Remarks to the Author):

The authors have answered/clarified my questions/concerns and added new experiments which resulted in significantly improved manuscript